# Humoral Immune Response Evaluation in Horses Vaccinated with Recombinant *Clostridium perfringens* Toxoids Alpha and Beta for 12 Months

**DOI:** 10.3390/toxins13080566

**Published:** 2021-08-13

**Authors:** Nayra F. Q. R. Freitas, Denis Y. Otaka, Cleideanny C. Galvão, Dayane M. de Almeida, Marcos R. A. Ferreira, Clóvis Moreira Júnior, Marina M. M. H. Hidalgo, Fabricio R. Conceição, Felipe M. Salvarani

**Affiliations:** 1Instituto de Medicina Veterinária, Universidade Federal do Pará, Castanhal CEP 68740-970, Brazil; nayraffreitas@gmail.com (N.F.Q.R.F.); otaka@veterinario.med.br (D.Y.O.); annymedvet@gmail.com (C.C.G.); dayanevet2016@gmail.com (D.M.d.A.); 2Centro de Desenvolvimento Tecnológico, Núcleo de Biotecnologia, Universidade Federal de Pelotas, Rio Grande do Sul CEP 96160-000, Brazil; marcosferreiravet@gmail.com (M.R.A.F.); clovismoreirajr@live.com (C.M.J.); fabricio.rochedo@ufpel.edu.br (F.R.C.); 3Faculdade de Veterinária, Universidade Federal de Pelotas, Rio Grande do Sul CEP 96160-000, Brazil; marina.haller@outlook.com

**Keywords:** recombinant alpha toxin, recombinant beta toxin, vaccine, myonecrosis, enterocolitis

## Abstract

In horses, *Clostridium perfringens* is associated with acute and fatal enterocolitis, which is caused by a beta toxin (CPB), and myonecrosis, which is caused by an alpha toxin (CPA). Although the most effective way to prevent these diseases is through vaccination, specific clostridial vaccines for horses against *C. perfringens* are not widely available. The aim of this study was to pioneer the immunization of horses with three different concentrations (100, 200 and 400 µg) of *C. perfringens* recombinant alpha (rCPA) and beta (rCPB) proteins, as well as to evaluate the humoral immune response over 360 days. Recombinant toxoids were developed and applied to 50 horses on days 0 and 30. Those vaccines attempted to stimulate the production of alpha antitoxin (anti-CPA) and beta antitoxin (anti-CPB), in addition to becoming innocuous, stable and sterile. There was a reduction in the level of neutralizing anti-CPA and anti-CPB antibodies following the 60th day; therefore, the concentrations of 200 and 400 µg capable of inducing a detectable humoral immune response were not determined until day 180. In practical terms, 200 µg is possibly the ideal concentration for use in the veterinary industry’s production of vaccines against the action of C. perfringens in equine species.

## 1. Introduction

Clostridium infections are a group of important domestic animal diseases caused by exotoxins produced by the bacteria of the genus *Clostridium.* These include toxin-mediated infections caused by *Clostridium perfringens*, which are responsible for enteric diseases and myonecrosis. These affect different livestock production systems, especially cattle, pigs and horses [1,2,3]. *C. perfringens* was reclassified into seven toxin types (A–G) that produce alpha (CPA), beta (CPB), epsilon (ETX), iota (ITX) and enterotoxin (CPE) toxins, all of which determine pathological conditions and clinical progression [2].

In horses, *C. perfringens* is associated with clinical presentations of acute myonecrosis that are caused by toxin type A or by the direct action of CPA. This then leads to hemorrhagic edema and gas formation in subcutaneous tissue, as well as adjacent muscles that crackle upon palpation [4,5,6] and severe cases of hemorrhagic enterocolitis in adult animals and neonatal foals [7,8,9,10], caused by a CPB-induced type C toxin infection. This disease is usually acute and super-acute, with a generally fatal outcome [4,9].

It is difficult to eradicate *C. perfringens* and any other species of the same genus because this bacterial agent is ubiquitous as a commensal microbiota in the gastrointestinal tract of healthy animals. It also has a high sporulation capacity, remaining viable for long periods within the environment [2]. Therefore, the most effective prophylaxis against these toxin infections is through vaccination [11]. However, the industrial production of toxoids containing *C. perfringens* antigens is costly and consists of several manufacturing steps. It also requires special care regarding biosafety for workers and the environment, and Clostridium produces low toxin levels in vitro [12]. In addition to all of this, to date, there have been no commercial vaccines available on the market against *C. perfringens* that are recommended for horses.

Over the last decade, studies involving the production and evaluation of clostridial recombinant vaccines have shown that these immunobiological agents are an important alternative to inducing humoral immune responses in farm animals, achieving higher neutralizing antibody titers when compared to conventional commercial vaccines [13,14,15,16,17]. Thus, the objective of this study was to serve as a pioneer in the use, evaluation and comparison of the humoral immune response duration in horses immunized with recombinant vaccines containing different concentrations (100, 200 and 400 µg) of *C. perfringens* recombinant alpha (rCPA) and beta (rCPB) proteins for 12 months.

## 2. Results

The sterility test indicated no microbiological growth after a 21-day observation period, demonstrating that there was no contamination during the formulation of the recombinant vaccines. In the innocuity test, no inoculated animals developed local or systemic adverse reactions.

The animals in the negative control group (G5) exhibited no anti-CPA and anti-CPB titers during the 360 days of the experiment. Figure 1 shows the mean CPA and CPB antitoxin titers induced by CV, RV1, RV2 and RV3 on days 60, 90, 120, 150 and 180, compared using the Bonferroni test (*p* < 0.001). Antibody titers decreased after day 60 in all vaccine groups. After day 210, no detectable antibody titers against *C. perfringens* CPA and CPB were found in any experimental group.

The percentage of animals presenting vaccine-induced seroconversion, and the mean levels of neutralizing anti-CPA and anti-CPB antibody titers for each vaccine group on days 60, 90, 120, 150 and 180, are shown in Table 1. Only RV2 and RV3 achieved the mean titer of at least 4 IU/mL for CPA antitoxin/ recommended by the USDA guidelines [18] and 10 IU/mL for CPB antitoxin, as required by Brazilian legislation, based on the European Pharmacopoeia [19].

Groups G2 and G3, vaccinated with the RV2 and RV3 vaccines, showed similar performance regarding seroconversion, with all immunized animals seroconverting and showing mean titers obtained for anti-CPA and anti-CPB from day 0 to day 180, with no significant differences (*p* = 0.4018). Only 50% of G1 horses vaccinated with RV1, demonstrated seroconversion for both recombinant toxins. In G4, inoculated with CV, only 30% of the animals seroconverted for CPA and 70% for CPB. The CV and RV1 vaccines induced mean neutralizing antibody titers only until day 120.

## 3. Discussion

Although toxin-induced infections caused by *C. perfringens* in horses were found to determine super-acute conditions, most of them were unresponsive to treatments [6,8]. However, there are still no CVs available that are specifically recommended for horses. In addition, commercial clostridial vaccines have a series of limitations in terms of their production process that may prevent homogeneity. In contrast, clostridial recombinant vaccines were shown to be a viable alternative for animal immunization [2,12], as previously demonstrated in different studies. Otaka et al. [20] compared the humoral immune response of vaccinated buffaloes to three different concentrations of recombinant proteins and a commercial botulism vaccine, reporting that higher and long-lasting neutralizing antibody levels were found in animals vaccinated with the recombinant vaccines, and that there was a direct relationship between the increased immune response longevity and higher concentrations of the recombinant proteins being evaluated.

The tested CV could not induce the mean titer levels required by the current Brazilian legislation in all vaccinated horses. This could be explained by the possibly low concentrations of effective antigens contained in the CV, resulting from the different stages of the detoxification process [12]. Another reason for this could be the lack of research proposing an adequate vaccination protocol against *C. perfringens* using CVs in horses to elicit a better humoral response. However, Moreira et al. [16] and Silva et. al. [21] demonstrated that the commercial clostridial vaccines evaluated were not able to stimulate humoral immune responses within the minimum levels required by legislation in all vaccinated animals, even when using vaccines specific to the species.

Many studies assessing recombinant clostridial vaccines that induce humoral immune response showed that these immunogens are an efficient alternative to CVs for several domestic species, such as pigs [13], cattle, sheep, goats [22] and buffalo [14]. To our knowledge, this study is the first to develop and test bivalent recombinant vaccines in horses over a one-year period and evaluate the duration of the immune response induced by different concentrations (100, 200 and 400 µg) of rCPA and rCPB recombinant proteins, using a CV for the control group. Similar to the aforementioned studies, our study reports not only that the recombinant toxoids can induce an immune response within the standards required by the current legislation, but also that the titrations found in horses vaccinated with RV2 and RV3 were significantly higher (*p* < 0.0001) when compared to animals vaccinated with the CV on days 60, 90, 120, 150 and 180. These results concur with Moreira et al. (2018) [16], Moreira et al. (2020) [17] and Otaka et al. [20], who found a significantly higher seroconversion rate in cattle and buffaloes vaccinated with *Clostridium botulinum* types C and D recombinant toxoids at 200 and 400 µg concentrations compared to the rate induced with a commercial toxoid.

The 200 µg (RV2) and 400 µg (RV3) concentrations of each of the recombinant toxoids administered to groups G2 and G3, respectively, induced an immune response against CPA and CPB until day 180, with mean titers above zero after the first vaccination. In turn, the CV and RV1 (100 µg) stimulated lower mean antibody titers that were detected only until day 120. The 200 µg (RV2) concentration of rCPA and rCPB was the lowest concentration capable of generating seroconversion in 100% of the vaccinated horses 60 days after the first vaccination. It also did not show statistically significant differences (*p* = 0.4018) between the induced titration for both toxoids compared to RV3 (400 µg) over the study period. The cost–benefit ratio shows that RV2 presented similar results to RV3, even though it contained half the concentration, and that 200 µg is possibly the ideal concentration to be used on an industrial scale to produce vaccines against toxin types A and C in horses.

After day 210, antibodies were not detected in any of the studied groups, which may have been a limitation of the immunodiagnostic technique; the test level used in serum neutralization detects no antibody titers below 4 IU/mL for anti-CPA, and 10 IU/mL for anti-CPB [15,23]. However, this could have also been related to vaccination protocols or even to the type of adjuvant used, which requires the development of new vaccine strategies that propose semiannual vaccination protocols, as well as the use and development of new adjuvant and immunostimulant molecules to maintain circulating neutralizing antibody levels for periods of up to one year. New studies should also be conducted to evaluate and modulate humoral immune responses via neutralizing antibody titration, and to stimulate the generation and quantification of memory cells. This is another important vaccination objective, because memory cells can survive for years and respond more quickly and effectively against antigens [24].

The results of this pioneering study show that recombinant vaccines at 200 and 400 µg concentrations induce a humoral immune response in horses and could be used as a preventive immunobiological agent against *C. perfringens* CPA- and CPB-toxin-induced infections. The 400-µg concentration was the most effective in the generation of higher and longer-lasting antibody levels. None of the recombinant vaccines used maintained detectable anti-CPA and anti-CPB titers throughout the year, indicating the need for further investigations into this topic to guide the development of better formulations, adjuvants and vaccine protocols for horses.

## 4. Materials and Methods

### 4.1. Ethics Committee

The study was conducted in accordance with the guidelines of the National Council for Animal Experimentation Control and approved by the Animal Ethics Committee of the Federal University under number 7310201016.

### 4.2. Recombinant Vaccines

The expression and production of rCPA and rCPB was performed according to the methodology described by Milach et al. [11] and Salvarani et al. [13]. The formulations for the three recombinant vaccines used rCPA and rCPB in 100, 200 and 400 µg concentrations (RV1, RV2 and RV3), adsorbed in an aluminum hydroxide suspension (2.5–3.5%Al (OH)_3_), and were continuously agitated at room temperature for 24 h [19]. Finally, 10 × 2-mL doses were obtained for each vaccine at 100 (RV1), 200 (RV2) and 400 µg (RV3) concentrations for each of the alpha and beta recombinant proteins.

### 4.3. Sterility and Safety Testing

To confirm the sterility of the recombinant vaccines produced, 0.5 mL of each vaccine formulation was transferred to four tubes that contained 20 mL of thioglycolate broth, and in four tubes that contained 20 mL of Sabouraud broth. Two thioglycolate broth tubes were incubated under anaerobic conditions, and the other Sabouraud and thioglycolate broth tubes were incubated under aerobic conditions. All tubes were maintained at 37 °C for 21 d, with daily readings [23].

For the innocuity test [20], two horses were inoculated via the intramuscular route with a vaccine formulation containing 800 µg of each of the recombinant toxoids, twice the highest tested concentration (400 µg), in order to evaluate the occurrence of local and systemic reactions for seven days.

### 4.4. Commercial Vaccine

A VISION 10^®^ (lot number 002/15, MERCK) commercial vaccine (CV) for cattle and pigs containing CPA and CPB *C. perfringens* toxoids was used and tested. The vaccine had no manufacturer specifications regarding the concentrations of each antigen to be used in the total formulation or per dose.

### 4.5. Animal Immunization and Titration of Neutralizing Antibodies

This study included 50 *Mangalarga*
*Marchador* horses of both sexes, aged 1 to 12 years, with no history of vaccination or *C. perfringens* alpha (anti-CPA) and beta antitoxin (anti -CPB) titers. The animals were randomly divided into 5 groups with 10 animals each: RV1, 100 µg (G1); RV2, 200 µg (G2); RV3, 400 µg (G3); CV (G4); negative controls (G5), which were administered sterile saline solution (NaCl 0.9%). The animals were kept in Guinea grass paddocks with water ad libitum along with mineral and protein supplementation. All animals received 2 separate 2-mL doses via intramuscular route on the neck, on days 0 and 28.

Blood samples were collected monthly from the jugular vein on days 60, 90, 120, 150, 180, 210, 240, 270, 300, 330 and 360. After clotting, the samples were centrifuged at 3000 rpm for 5 min to obtain serum, which was kept at −20 °C for subsequent processing.

A neutralization assay was used for serum titration. We employed the methodology proposed by the United States Department of Agriculture (USDA) [18] to detect CPA antitoxin, put forth by the European Pharmacopoeia [19], in order to detect CPB antitoxin. The neutralization titer was calculated according to the method developed by Reed and Muench [25] and expressed in international units per milliliter (IU/mL).

### 4.6. Statistical Analysis

Analysis of variance and the Bonferroni post-hoc test were used to identify statistically significant differences in antibody titers between groups using Statview statistical software, version 5.0.0.0. Only *p* < 0.001 was considered statistically significant.

## Figures and Tables

**Figure 1 toxins-13-00566-f001:**
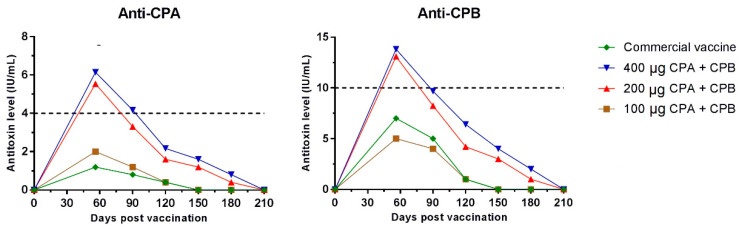
Mean alpha (anti-CPA) and beta antitoxin (anti-CPB) titers in horses immunized with a commercial vaccine (CV) and with the three recombinant vaccines (RV1, RV2 and RV3) on days 60, 90, 120, 150 and 180 after the first vaccination.

**Table 1 toxins-13-00566-t001:** The duration of alpha (anti-CPA) and beta antitoxin (anti-CPB) titers in horses immunized with a CV and with the three recombinant vaccines (100, 200 and 400 µg) on days 60, 90, 120, 150 and 180 after the first vaccination.

	DAFV ^1^		Anti-CPA Mean Titer (IU/mL) ± SD ^2^						Anti-CPB Mean Titer (IU/mL) ± SD ^2^				
Formulations		60	90	120	150	180	SR ^3^	60	90	120	150	180	SR ^3^
RV1 (G1)		2.0 ^±2.1^	1.2 ^±1.9^	0.4 ^±1.2^	0	0	50%	5.0 ^±5.2^	4.0 ^±5.1^	1.0 ^±3.1^	0	0	50%
RV2 (G2)		5.5 ^±1.1^	3.3 ^±2.4^	1.6 ^±2.0^	1.2 ^±1.9^	0.4 ^±1.2^	100%	13.1 ^±2.3^	8.2 ^±5.8^	4.2 ^±5.4^	3.0 ^±4.8^	1.0 ^±3.1^	100%
RV3 (G3)		6.1 ^±1.1^	4.1 ^±2.3^	2.1 ^±2.3^	1.6 ^±2.0^	0.8 ^±1.6^	100%	13.8 ^±2.3^	9.6 ^±5.3^	6.4 ^±5.5^	4.0 ^±5.1^	2.0 ^±4.2^	100%
CV (G4)		1.2 ^±1.9^	0.8 ^±1.7^	0.4 ^±1.2^	0	0	30%	7.0 ^±4.8^	5.0 ^±5.2^	1.0 ^±3.1^	0	0	70%

**DAFV ^1^** days after the first vaccination; **SD ^2^** Standard deviation; **SR ^3^** Seroconversion rate.

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
