# Peer review of "Humoral Immune Response Evaluation in Horses Vaccinated with Recombinant Clostridium perfringens Toxoids Alpha and Beta for 12 Months"

_toxins, 2021, doi:10.3390/toxins13080566_

Round 1
Reviewer 1 Report
This article is an interesting paper that the administration of CPA and CPB toxoid increased the antibody titer in horses. Due to the problem of serum sickness, it is not recommended to use commercial antitoxins in horses. The authors are analyzing the protective effect of vaccines using recombinant toxins over time.
<Comments>
1) The antibody titer peaks 60 days after vaccination. How long do you expect the vaccine to be effective in vivo?
2) Change the unit in Figure 1 to microgram.
Author Response
Point 1 : The antibody titer peaks 60 days after vaccination. How long do you expect the vaccine to be effective in vivo?
Response 1: Antibody peaks at 60 days after the 1st vaccination, but levels were detected until day 180, which time of six months we expect a vaccine to be effective in vivo. However, we know that vaccination has individual variation and that a vaccinated animal is not 100% protected. The experiment demonstrates that the ideal would be a revaccination at 180 days and not at 360 days as established in the vaccine protocols.
Point 2: Change the unit in Figure 1 to microgram
Response 2: Changed the unit to microgram.
Reviewer 2 Report
This manuscript describes the results of immunization of horses with recombinant CPE and beta toxins from C. perfringens. The inoculations with 200 micrograms and 400 micrograms performed better in eliciting a serum response than those with 100 micrograms or a commercial vaccine of unknown constituents. Unfortunately, the antibody response lasted only 180 days and apparently did not elicit long term protection. As noted by the authors, these results are preliminary in nature and more work needs to be done on selecting better adjuvants and testing for the presence of memory B cells.
The text does need extensive editing for proper English syntax.
Author Response
Point 1: This manuscript describes the results of immunization of horses with recombinant CPE and beta toxins from C. perfringens. The inoculations with 200 micrograms and 400 micrograms performed better in eliciting a serum response than those with 100 micrograms or a commercial vaccine of unknown constituents. Unfortunately, the antibody response lasted only 180 days and apparently did not elicit long term protection. As noted by the authors, these results are preliminary in nature and more work needs to be done on selecting better adjuvants and testing for the presence of memory B cells.
Response 1: We agree with the reviewer that a revaccination of the animals at 180 days would be necessary for the level of serum antibodies to remain detectable. Today, vaccine protocols define only annual revaccination. We also agree that new adjuvants, new recombinant antigens, new vaccine protocols and the measurement of memory B cells should be the basis for further studies. The important and differential of this work was to demonstrate that the recombinant proteins used in the work were immunogenic and safe for vaccinated horses, which had never been done and described in the literature.
Point 2: The text does need extensive editing for proper English syntax.
Response 2: The article will be sent for editing in English as recommended by the reviewer and in the companies suggested by Toxins. Please see the attachment.

Round 2
Reviewer 1 Report
The manuscript is properly proofread and the comments are properly answered.